# Refugees Welcome? Online Hate Speech and Sentiments in Twitter in Spain during the Reception of the Boat Aquarius

**Carlos Arcila-Calderón** [1,*], **David Blanco-Herrero** [1], **Maximiliano Frías-Vázquez** [1] and **Francisco Seoane-Pérez** [2]

1   Facultad de Ciencias Sociales, Campus Unamuno, University of Salamanca, 37007 Salamanca, Spain; david.blanco.herrero@usal.es (D.B.-H.); maxfrias@usal.es (M.F.-V.)
2   Departamento de Periodismo y Comunicación, University Carlos III of Madrid, 28903 Madrid, Spain; fseoane@hum.uc3m.es
*   Correspondence: carcila@usal.es

**Abstract:** High-profile events can trigger expressions of hate speech online, which in turn modifies attitudes and offline behavior towards stigmatized groups. This paper addresses the first path of this process using manual and computational methods to analyze the stream of Twitter messages in Spanish around the boat Aquarius (n = 24,254) before and after the announcement of the Spanish government to welcome the boat in June 2018, a milestone for asylum seekers acceptance in the EU and an event that was highly covered by media. It was observed that most of the messages were related to a few topics and had a generally positive sentiment, although a significant part of messages expressed rejection or hate—often supported by stereotypes and lies—towards refugees and migrants and towards politicians. These expressions grew after the announcement of hosting the boat, although the general sentiment of the messages became more positive. We discuss the theoretical, practical, and methodological implications of the study, and acknowledge limitations referred to the examined timeframe and to the preliminary condition of the conclusions.

**Keywords:** aquarius boat; immigration; hate speech; politicians; refugees and asylum seekers; sentiment analysis; topic modelling; twitter

## 1. Introduction

Since the Crisis of Refugees of the Mediterranean in 2015, Southern European countries have gained relevance in the reception of refugees, asylum seekers, and migrants that either remain in them or continue to wealthier countries. Although Italy and Greece have been the arrival countries for most migrants, the eventual closure of the migration routes from Turkey to Greece and from Lybia to Italy turned Spain into the country with more Mediterranean arrivals in 2018, with 57,250 Africans reaching Spanish soil according to the International Organization for Migration [1]. The biggest single intake was that of the boat Aquarius, which carried 630 migrants and arrived at the port of Valencia in June 2018 after being rejected by other European countries. The announcement by the Spanish government on 11 June 2018 that the country would welcome the drifting boat was the first high-profile political move of the new Socialist executive, who had reached power 10 days after a successful motion of censure in the national Parliament.

In just 1 week of that month, the case of Aquarius generated a large amount of interactions in Twitter (~24,000), making this social medium a unique space to monitor and analyze the public opinion towards refugees. This is an example of a high-profile real-life event that has led to a strong activity in social media, something that has been often studied in the academia, given that these events can act as drivers in the expression of either support or rejection, giving also space for different forms of hate speech. Miró-Llinares and Rodríguez-Sala [2] studied Twitter's conversation after the attacks against the headquarters of Charlie Hebdo in Paris in 2015, while Burnap and Williams [3], and Williams and Burnap [4] did something similar after the murder of Drummer Lee Rigby in

London in 2013. The relevance of studying these cases is increased by the fact that the hate spiral produced after these events also led to episodes of verbal violence and attacks against the honor and the dignity of specific individuals, which is already a crime by itself [5].

Hate speech generally refers to all forms of discourse that deteriorate the image of a person or a group of individuals based on their inherent or acquired condition. This includes explicit hate messages, as well as subtler narratives by which the image of groups is despised with the goal of exercising social control. The pioneering scholar connecting hate speech and communication theory was Calvert [6], who drew from James Carey's famous distinction between "communication as transmission" and "communication as culture" to ponder on the difficulties of measuring the actual effect of hateful speech on violent behavior. Jurors are always looking for evidence following the transmission model, linking threats to actions in the short term, whereas a cultural perspective invites us to think about the indirect consequences of hateful discourse as a breeding ground to justify discrimination in the long term. Regardless of one's preference for the transmission on the cultural model, Calvert's insight remains valid in this digital age: Any strategy that seeks to understand and fight hate speech must include a communication approach. Additionally, recent studies have found a link between online hate speech and the increase of hate crime against this vulnerable group [7], supporting a broader but still inconclusive scientific literature on predicting real world behaviors based on media texts [8–10], and also supporting the fact that combating hate speech can decrease the number on hate crimes.

With this in mind, the study will use text mining, or text analytics, not only for the need of overcoming the limitations of surveys in racism—or xenophobia-related issues [11], but also since the access to unstructured data, such as the one of Twitter, provides unfiltered and immediate information in a particular moment in a way that surveys or other studies cannot. Text mining refers to a set of techniques that are gaining relevance in different fields of the academia, from finance [12] to social impact [13], including politics [14] and anti-immigration sentiments [15], the topics of this study. From these techniques, which offer a way to extract knowledge from textual information using big data [16], sentiment analysis and topic modelling, the two that will be used in this study, are among the most common ones, both with a growing presence, according to the systematic review of literature on text mining on Twitter conducted by Karami et al. [17].

With all this, the goal of this paper is to understand the features of refugee-related messages in Twitter around a high-profile event, such as the announcement of Spanish Prime Minister Pedro Sánchez to welcome the ship Aquarius, as well as identifying the expressions of hate and the role of the event in the variation of hatred contents towards refugees. This will allow us to fill in a still existing knowledge gap on the feelings of the Spanish population towards refugees and immigrants, not only around this specific high-profile event, but also in the context immediately before the arrival of Vox to Spanish institutions in December 2018, which made anti-immigration a more relevant topic in the Spanish political discourse [18]. This way, the study will complement previous works in the field and try to observe how and around what topics the anti-immigration discourse in Spain is built [15,18,19] and, more in general, how refugees are depicted in Twitter [20,21]. Moreover, the article introduces a novelty in its approach to hate speech against politicians, even if they are not a vulnerable group that demands protection, one of the most common reasons behind hateful contents is political ideology. Therefore, approaching these types of contents, something very seldomly done in the past, will offer a basis for future works analyzing ideologically motivated hatred.

In the next sections we justify our research questions (Section 1.1.1.) and hypothesis (Section 1.1.2.), describe the manual and computational methods used in the research (Section 2), explain our findings (Sections 3 and 4) and, finally, present our conclusions (Section 4).

## 1.1. Online Hate Speech

The two legal schools of thought regarding hate speech somehow mimic the distinction between the transmission and cultural approaches in communication theory. The American model, grounded on the "marketplace of ideas", only prohibits hateful speech if there is a clear and present danger of physical violence. The European approach, borrowing from Germany's post-war "militant democracy", is far more coercive, concerning itself with the hateful expressions that might lead to a certain climate of opinion, even if no immediate danger can be devised [22].

Hate speech must be differentiated from mere offensive language, or an insult. Hate speech has a clear social dimension, as it is targeted to a specific group marked by a distinctive feature, be it race, religion, ideology, or sexual orientation, etc. The European school of legal thought on dangerous speech is clearly marked by the experience of genocide. In its first relevant decision on the topic, the Council of Europe issued a Recommendation in 1997 (97/20, 30 October) asking its constituent States to act against "all forms of expression which spread, incite, promote or justify racial hatred, xenophobia, anti-Semitism or other forms of hatred based on intolerance, including: intolerance expressed by aggressive nationalism and ethnocentrism, discrimination and hostility against minorities, migrants and people of immigrant origin". After 7 years of negotiations among its member states [22], the Council of the European Union agreed in 2008 on a definition of *illegal* hate speech as "publicly inciting to violence or hatred directed against a group of persons or a member of such a group defined by reference to race, colour, religion, descent or national or ethnic origin" [23].

To better understand these definitions, it should be noted that online hate speech is a specific type of hate which is part of the behavior of the so-called cyber hate. In a context in which social media, such as Facebook or Twitter, and new digital media have allowed a faster and broader spreading of these contents, as well as a greater visibility, online hate speech becomes more dangerous and potentially harmful than previous expressions of hate. This has led to a growing interest of studies dealing with the problem of online hate speech, such as Müller and Schwarz [7], who showed how right-wing anti-refugee sentiment on Facebook predicts violent crimes against refugees; Lucas [24], who collected different international experiences of monitoring and mapping online hate speech or Burnap and Williams [3], who used machine classification to model hate speech online.

Moretón Toquero [25] discusses that, following the principle of minimum intervention, cyber hate specifically applies, from a legal point of view, to the most harmful expressions which are regulated in the Criminal Code, and that freedom of speech tends to protect everything else. However, legally punishable forms of hate speech are just the most visible and harmful part of verbal rejection. Contrada et al. [26] considered verbal rejection as the most basic form of discrimination, whereas Brown [27] defended that rejection towards the other can go from verbal rejection to genocide. In our field, previous studies [20] have pointed out how the rejection of immigrants and the demand of their expulsion or the prohibition of their entrance, especially by prominent figures and opinion leaders, can lead to an increase of hate speech. Therefore, in order to understand hate speech and rejection towards immigrants from a broad perspective, with the goal of tackling the problem from its basis, in this study we will not only try to identify explicit expressions of hate speech, as defined by law, but also expressions of rejection which do not fit into that classification but can potentially lead to others that do.

### 1.1.1. Refugees as a Subject in Social Media

The predominance of social media and the new dimension of hate speech, thanks to them, has led to an increase in the regulation and legislation of hate speech online [28,29]. However, the popularity of online social media keeps generating new challenges, such as the spread of fake news [30], the use of disinformation for promoting rejection against refugees [31] or the increase in the presence of hate speech [32] despite all efforts. However, social media have also become relevant to gauge the public opinion of societies, including

the presence of racism [33] or the representation of refugees [21]. As a result, authors such as Ross et al. [34] see a great interest in the identification of hate speech in social media in connection to real-life events, such as, in our case, the arrival of a boat to the coast of Spain in the context of the crisis of refugees in the Mediterranean.

More specifically, Twitter has been frequently used to discover the political and public discourse of a society on different sociopolitical topics [35], including immigration-related topics [20,36]. The interest of the academia in Twitter comes from the condition of digital tribune of this micro-blogging medium, in which users express organically and freely their thoughts and opinions feeling as they are talking to a captive audience [33]. In addition, although it is not a representative platform of all society, its popularity, the possibility of making contents viral using hashtags or mentions, and the ease and speed of its communication, make it of great interest for our analysis.

Previous research [37] shows the importance of analyzing the predominant sentiment of the discourse of social media due to its capacity to provide information on the context and the possible reactions towards any minority group. Sentiment analysis has also been related to the presence of hateful content [38]. Therefore, we present the following research question:

Which is the predominant sentiment of messages in Twitter in Spanish referring to the arrival of the boat Aquarius? (RQ1).

The Statistic System of Criminality (SEC) of the Ministry of the Interior of Spain refers to eight motivations or prejudices that can lead to hate expressions: Racism/Xenophobia; Sexual orientation or identity; Religious praxis or beliefs; Ideology; Disability; Gender, Antisemitism; and Aporophobia. In the same line, Olteanu et al. [39], who analyzed the discourse of xenophobic hatred in Twitter, discovered that there are six potential publics towards whom this hateful discourse is usually aimed: "Muslims and Islam; Religious groups: unspecified, any religion except Islam; Arabs, Orientals or North Africans: offspring without reference to religion; Ethnic groups or groups of foreign ancestry: unspecified, any foreign ancestry, except Arabic; Immigrants, refugees or foreigners in general: without indicating a specific religion or offspring; Other groups of nonimmigrants: according to, for example, gender, sexual orientation, appearance, disability or age" (p. 5). Following these classifications, we will focus here on refugees (who can be a target of discrimination and rejection in various of the previously mentioned categories) to pose the following research question:

To what extent is hate speech towards refugees present in the messages in Twitter in Spanish referring to the arrival of the boat Aquarius and how does the presence of hate speech relate with the predominant sentiment in those messages? (RQ2).

In order to understand the reasons that explain those feelings, and following Entman [39], it is important to know what the topics present in the messages are and under which frame they are presented, which will allow us to know what are the problems, the causes or the risks that could explain the presence of hate speech towards refugees and migrants, but also which are the possible solutions against these narratives. Therefore, this research question will be complemented with this secondary one:

What are the main topics behind the messages in Twitter in Spain with hate speech against refugees? (RQ2.1).

Olteanu et al. [40] suggested that, although most messages containing hate are aimed towards minority groups—in this case, those would be the migrants and refugees, it is also possible to find messages that directly attack non-vulnerable groups, such as political parties, media organizations or senior profiles of politicians and journalists. Therefore, with the goal of broadening the lens about who could become a victim of hateful expressions in social media and to discover what is the sentiment of the hateful messages aimed at each public, we present the following research question:

To what extent is hate speech towards politicians present in the messages in Twitter in Spanish referring to the arrival of the boat Aquarius and how does the presence of hate speech relate with the predominant sentiment in those messages? (RQ3).

As in RQ2, we can dig into the problems, the causes or the risks that could explain the presence of hate speech or of negative sentiments towards this group. Therefore, this research question will be complemented with this secondary question:

What are the main topics behind the messages in Twitter in Spain with hate speech against politicians? (RQ3.1).

### 1.1.2. High-Profile Events and Hate Speech

It can be seen how messages spread by specialized or organized groups, and including those published with an appearance of news [31], on the Internet are means of transport for hate, increasing the presence of hate in the public discourse. However, some authors [3,41–43] defended that online hate speech can be also intensified in social media after high-impact events, such as terrorist attacks or news with the presence of disadvantaged or denigrated groups of population, and may be the springboards to incidents and targeted attacks with similar characteristics. Sometimes, even apparently unrelated events can be used as an excuse for an increase in hate speech, often in connection with disinformation or fake news campaigns [44].

Furthermore, previous studies have found that certain events can affect and modify the public perception of refugees and migrants [45–48] from the situation prior to a high impact event to the response after that. In the case of the boat Aquarius, the turning point took place when the Government officially announced that Spain would welcome the boat, therefore, we present the following hypothesis:

The average of hateful comments and negative sentiments in the messages in Twitter in Spanish referring to the arrival of the boat Aquarius increased after the official announcement of welcoming the boat in Spain (H1).

## 2. Materials and Methods

### 2.1. Sample

Using the Stream API of Twitter [49], we collected those messages that declared language in Spanish and included "#Aquarius" in the field *text* from 7:27 a.m. 8 June to 11:54 p.m. 17 June 2018 (N = 26,236). After removing 1983 messages that referred to other topics or were written in other languages, the final sample was of n = 24,254 messages. These contents were downloaded and analyzed in the servers of the Supercomputing Centre of Castille and León (Scayle).

### 2.2. Measures

Five measures were used in this study: Three of them were manually analyzed, whereas the last two were computationally analyzed. These measures were:

1. Date and time: It shows the date and the time in which the message was produced. This measure allowed the division of the tweets between those produced before the announcement of the Spanish Prime Minister, Pedro Sánchez, of receiving the boat Aquarius in the harbor of Valencia (0) and those produced afterwards (1). This announcement took place on 11 June at around 12.30 p.m. (Madrid's time zone, GMT+1; the moment will be marked in the set of tweets with the following tweet by @AlejandroCancho, posted at 12.30.41: "*#ÚltimaHora - El gobierno de #España acogerá el #Aquarius con 629 migrantes que hasta ahora no tenían donde atracar el barco*"), and this division allowed us to answer H1.

2. Presence of hateful expressions: It shows whether the message is of a hateful nature (1; as previously discussed, this category includes messages with explicit forms of hate speech, but also other expressions of rejection that could lead to it) or not (0; in this case, it includes messages showing support, but also those of an informative nature). If a message simultaneously showed support to a group but rejection to another, the predominant feeling was determined by the coder, so that all messages had only one category.

3. The target of the hateful expression: In order to determine the target of the hateful expressions, two complementary measures were created: The first one showed the presence of hate speech against refugees (1) or not (0, including those tweets with hate against

politicians, but also those that had no hate or rejection); the second one showed the presence of hate speech against politicians (1) or not (0, including those tweets with hate against refugees, but also those that had no hate or rejection). Given that all the collected messages had the hashtag "#Aquarius" and that only those referring to the arrival of the boat were considered for the analysis, they all referred to one of these groups. In order to simplify the analysis, we determined that tweets could only show rejection towards one of the two groups—refugees or politicians. In the cases in which hate or rejection could be found towards both groups, coders were instructed to select the predominant one. It should be also highlighted here that we are not differentiating between politicians of different nationalities, ideologies and, especially in this analysis, with different approaches towards migration, given that none of them can be considered vulnerable groups. Similarly, the expressions towards refugees or migrants were studied together since, although some were aimed at the people rescued by the boat Aquarius and other at immigrants in general, they all shared the condition of immigrants and, therefore, of vulnerable members of the *exogroup*, which are more exposed to prejudice and rejection [50,51].

These first three measures were manually analyzed by one previously trained coder within a period of 3 months, from mid-September 2018 to mid-December 2018. In order to test the reliability of this analysis, a secondary coder content analyzed a random subsample of 1293 messages (~5% of the total sample of 26,236 tweets, so that we could also test the reliability of the removal of tweets that did not belong to the sample) during the month of February 2019. Inter-coders reliability tests were conducted for measures 2 and 3, showing in both cases adequate values: $K\alpha = 0.709$ for the presence of hateful expressions and $K\alpha = 0.854$ for the target of the hateful expression. In addition to the presence (1) or not (0) of hate speech in the messages, a third category (99) was included in this measure, so that we could test the agreement of the coders regarding the removal of the messages which would not be of interest in the study, for being in other languages or for referring to unrelated topics. This means that for the inter-coders reliability test, prior to the removal of unrelated tweets for the analysis and to the analysis itself, measure 2 –presence of hate speech in the message– had three categories, although it only had two in the analysis –given that the removed tweets were already out of the sample.

In addition to the manual study of the tweets, we ran the automated sentiment analysis and topic modelling. The two measures for these analyses were:

4. Sentiment of the tweet: This automated analysis was based on a lexicon or dictionary, using *SentiStrength*, an open-source tool developed by Thelwall et al. [52] and validated in Spanish by Vilares, Thelwall, and Alonso [53]. This software rates the relevance and presence of negative words (from −1, being not negative, to −5, being extremely negative) and positive words (from +1, being not positive, to +5, being extremely positive) for every tweet. The sum of these two values indicates the general emotions of the tweet in terms of language (*Language Sentiment*) and it can be from −4 (+1, −5; clear predominance of negative words) to 4 (+5, −1; clear predominance of positive words).

5. Detected topics in the hateful messages: With the goal of discovering the main topics behind the messages, we applied unsupervised machine learning to the content of the messages. Specifically, we conducted topic modelling using Latent Dirichlet Allocation (LDA), which is the most commonly used algorithm for this unsupervised task [54]. Topic modelling is employed to identify topics in large sets of documents [55], in this case, a large set of tweets. We estimated the optimal number of topics based on *internal coherence* values (Figure 1) for messages with hate speech against refugees and messages with hate speech against politicians. This optimal number was determined using the Elbow method, which leads us to choose the number of clusters at which the variability abruptly decreases, creating an "elbow" in the slope. The selection of one "elbow" or another depends on the adequate number of topics that can be considered useful for the study. In this case, as it can be seen in the left part of Figure 1, the optimal number of topics for the 2397 tweets that expressed hate against refugees and migrants were 6 (*X*-axis), with an internal coherence of −7.54 (*Y*-axis; the internal coherence value for 0 to 15 topics would be, in the case of hate

against refugees: −3.62, −3.62, −4.47, −6.31, −5.42, −8.07, −7.54, −8.75, −9.75, −10.00, −10.02, −10.41, −11.53, −11.94, −12.61, −12.93), while the optimal number of topics for the 4146 tweets that expressed hate against politicians were 4 (*X*-axis), with an internal coherence of -6.48 (*Y*-axis; the internal coherence value for 0 to 15 topics would be, in the case of hate against politicians: −3.92, −3.92, −4.16, −6.19, −6.48, −7.81, −10.44, −10.24, −10.00, −12.09, −11.24, −11.50, −11.88, −12.37, −13.18, −12.76). In other words, the internal coherence value is the measure that orientates the optimal number of topics, which means that the researchers should later obtain a descriptive label for each topic attending to the main terms included in each of them (the standard number of terms is ten).

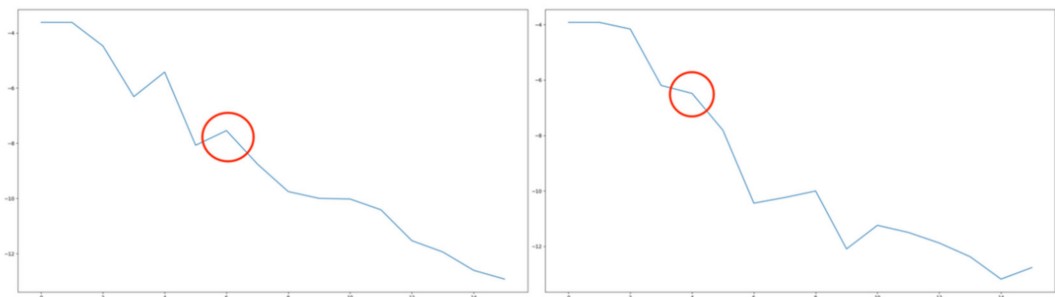

**Figure 1.** Optimal number of topics for the messages with hate against refugees (**left**) and politicians (**right**). The *X*-axes represent the number of topics and the Y-axes represent the value of the internal coherence.

After selecting the optimal number of topics for each group, we interactively conducted topic modelling over the two sets of messages. At each execution of LDA we obtained a list of words (10) that characterized every detected topic. By creating and updating a set of stop words, we manually removed the words of the initial list that did not contribute to generate a logic label of the topics (such as articles and other meaningless particles, but also variations of the verb "to be", for example), and repeated this process several times until obtaining a final set of relevant words with enough meaning to create the labels. The final amount of stop words was 386: 313 predefined words from the nltk package plus 73 ad-hoc terms.

It should be noted that this model, known as Clean Model, was selected after having tested others (Naïve, tf-idf, and tf-idt+trimming of extreme words) that did not offer the same coherence values, according to the *u-mass* statistic. These models were tested once all capital letters, all punctuation signs, and all html tags (boiler plates) were removed in a preprocessing phase.

### 2.3. Software Used

All the contents were downloaded in the JSON format, which allows the execution of filters by username, language, date, geographic location, etc. In this particular investigation, only the text contained in the messages was analyzed, using the publication date as an organizing feature, so the contents were cleaned up to remove the irrelevant information. The contents were later transformed into a data frame for subsequent automated analysis using Python, specifically, the Gensim library and the Latent Dirichlet Allocation (LDA) algorithm in order to discover the underlying issues within the sample, as well as Java and the *SentiStrengt* library in order to identify the emotions within the tweets.

Finally, IBM's SPSS program was used to perform the descriptive statistical analysis and inferential statistical analysis (including Student's *t*-tests for independent variables and correlations). Krippendorff's Alpha macro was also used to measure the reliability of the aforementioned inter-coders test.

### 3. Results

In the first place, we found that 27% of the messages of the final sample (6543) included expressions of hate or rejection, while 73% (17,711 messages) were not hateful messages.

Seeking to answer RQ1, we observed that the general feeling of the sample of tweets was slightly positive (M = 0.018; SD = 1.643), with a small predominance of positive (M = 1.960; SD = 1.139) rather than negative words (M = −1.940; SD = 1.280). These values are summarized in Table 1, together with those that will allow us to answer RQ2 and RQ3.

**Table 1.** Summary of frequencies and mean values.

|  | Frequency | Percentage | Sentiment | Positive Words | Negative Words |
|---|---|---|---|---|---|
| Non-hateful messages | 17,711 | 73% | 0.067 | 1.950 | −1.880 |
| Hateful messages | 6543 | 27% | −0.114 | 1.980 | −2.090 |
| Hate against refugees | 2397 | 9.9% | −0.073 | 1.960 | −2.040 |
| Hate against politicians | 4146 | 17.1% | −0.137 | 1.980 | −2.120 |
| Final sample | 24,254 | 100% | 0.018 | 1.960 | −1.940 |

Source: The authors.

We found a significant, although small, negative correlation between the presence of hate speech and the sentiment [r = −0.49; $p < 0.001$], meaning that the presence of hate speech correlates with a greater negativity of the message. More specifically, we can see how the presence of hate speech does not show a significant correlation with the relevance of the positive words, but it does with the relevance of the negative words [r = −0.73; $p < 0.001$] (we must remember at this point that the relevance of negative words grows when the values go farther from −1 and closer to −5).

Focusing now on the hateful or rejection messages aimed at refugees, we can answer RQ2 by saying that 2397 messages (9.9% of the total sample, and 36.6% of the hateful ones) showed hate or rejection against refugees or migrants, either those in the boat Aquarius or other immigrants in general. In these messages, the general feeling was slightly negative (M = −0.073; SD = 1.690), with a predominance of negative (M = −2.040; SD = 1.312) rather than positive words (M = 1.960; SD = 1.138).

We can find a small but significant negative correlation between the sentiment of the tweet and the presence of hate speech expression towards refugees [r = −0.18; $p < 0.01$], which means that the presence of hate or rejection against this target correlates with a stronger negative feeling in the text. More specifically, we could observe a significant correlation between the presence of hate speech against refugees and the importance of negative words in the messages [r = −0.26; $p < 0.001$]. This correlation shows that a greater relevance of negative words can be found in those texts with hate speech expressions against refugees and migrants. There was no significant correlation between the relevance of positive words and the presence of hate speech against these groups.

For a more detailed analysis of the topics present in these messages, we used LDA to answer RQ2.1. Table 2 shows the six topics (with each word and respective weight) that were found within these hateful messages against refugees and Figure 2 shows an example of the distribution of topics and the list of words of the first topic.

In general, it seems that the problem is not so much welcoming this particular boat and the asylum seekers that travel in it, but the further consequences of this decision. This way, we can see how the most relevant aspect seems to be the fear of the pull effect that hosting the boat could have on the arrival of new immigrants to Spain. This relates also with the fear of having terrorists among the immigrants and with the economic burden of what a large number of immigrants would mean for the State and the Spanish citizens.

Similarly to the previous research question, we can answer RQ3 by saying that a total of 4146 messages included expressions of hate or rejection against politicians. That equals 17.1% of the total sample of studied tweets and 63.4% of the contents with hate speech. The feeling of these tweets was rather negative (M = −0.137; SD = 1.673), with a stronger relevance of negative (M = −2.120; SD = 1.341) rather than positive words (M = 1.980; SD = 1.122).

**Table 2.** Topics of the messages with hate speech against refugees.

| Topic | Most Common Words and Their Frequency within the Topic (the Underlined Words are the Most Determinant for the Labelling of the Topic) |
|---|---|
| Pull effect and consequences | "immigrants" (0.008) + "spain" (0.007) + "boat" (0.005) + "<u>effect</u>" (0.005) + "<u>pull</u>" (0.004) + "goes" (0.004) + "valencia" (0.004) + "<u>consequences</u>" (0.004) + "europe" (0.004) + "<u>concentration</u>" (0.004) |
| Pull effect and not welcoming "illegals" | "immigrants" (0.013) + "people" (0.008) + "spain" (0.008) + "<u>pull</u>" (0.007) + "<u>effect</u>" (0.007) + "illegal" (0.007) + "<u>protectyourborders</u>" (0.007) + "spaniards" (0.006) + "harbor" (0.006) + "<u>aquariusnotwelcome</u>" (0.006) |
| Not welcoming and terrorism | "go" (0.008) + "spain" (0.008) + "people" (0.006) + "<u>aquariusnotwelcome</u>" (0.005) + "country" (0.005) + "<u>boko</u>" (0.005) + "<u>haram</u>" (0.005) + "immigrants" (0.005) + "boat" (0.004) + "have" (0.004) |
| Smugglers and NGOs | "immigrants" (0.023) + "spain" (0.010) + "spaniards" + "<u>come</u>" (0.008) + "<u>mafias</u>" (0.008) + "people" (0.007) + "go" (0.006) + "illegal" (0.006) + "<u>ngos</u>" (0.006) + "boat" (0.006) |
| Money and jobs | "refugees" (0.011) + "immigrants" (0.009) + "<u>pay</u>" (0.007) + "spain" (0.007) + "countries" (0.005) + "boat" (0.005) + "spaniards" (0.005) + "people" (0.005) + "<u>solution</u>" (0.005) + "<u>work</u>" (0.005) |
| Entrance to Europe | "spain" (0.018) + "immigrants" (0.009) + "europe" (0.006) + "boat" (0.006) + "valencia" (0.005) + "spaniards" (0.005) + "government" (0.005) + "immigration" (0.005) + "<u>north</u>" (0.004) + "<u>millions</u>" (0.005) |

Source: The authors. For the original Spanish terms, see Table A1 in Appendix A.

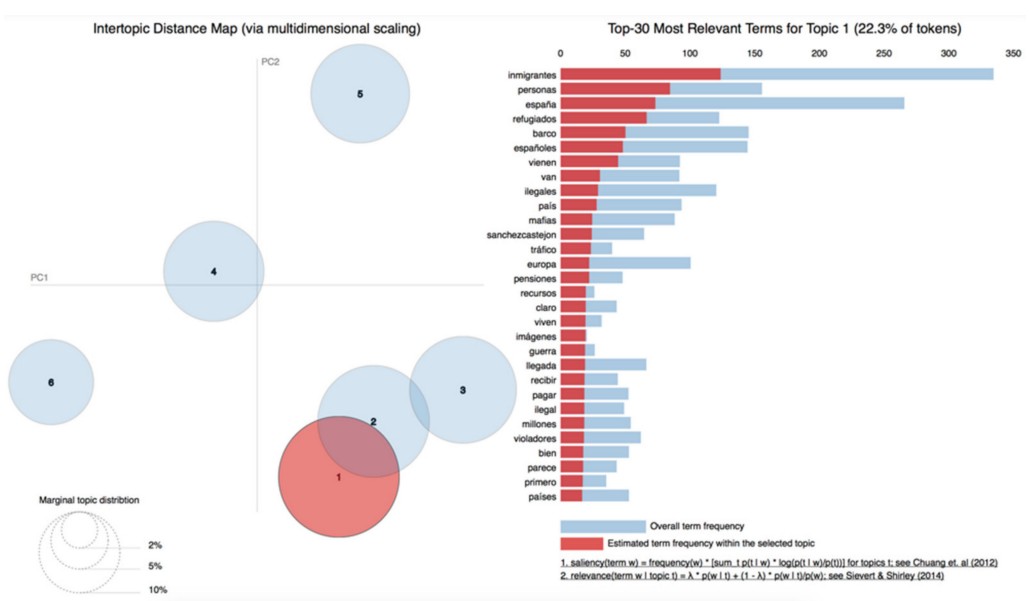

**Figure 2.** Topics within the hateful messages against refugees and list of words of the first topic.

The results also showed a statistically significant correlation between the presence of hate speech against politicians and the sentiment of the language of the message [r = −0.043; $p < 0.001$], which means that the presence of hate speech towards politicians was associated with a stronger relevance of negative words. This result is complemented by the statistically significant correlation between the presence of hateful contents aimed at politicians and the importance of negative words [r = −0.065; $p < 0.001$] in the message. This means that hate speech is more frequent in texts with a larger presence of negative words. However, no statistically significant correlation between the presence of hate speech against politicians and the importance of positive words in the message [r = 0.011; $p = 0.074$] was found. In other words, and similar to what was observed when analyzing the presence of hate speech against immigrants, a greater relevance of positive words in the message was not linked with the presence of hate speech.

Looking more in detail into the content of the messages, we can answer RQ3.1 by showing the four most relevant topics discovered among the tweets with hate speech against politicians (see Table 3 and Figure 3):

**Table 3.** Topics of the messages with hate speech against politicians.

| Topic | Most Common Words and Their Frequency within the Topic (the Underlined Words are the Most Determinant for the Labelling of the Topic) |
|---|---|
| Shame of politicians | "immigrants" (0.006) + "urdangarín" (0.005) + "people" (0.005) + "europe" (0.005) + "government" (0.004) + "pp" (0.004) + "italy" (0.004) + "<u>shame</u>" (0.004) + "<u>hypocrisy</u>" (0.004) + "open" (0.003) |
| Media coverage | "valencia" (0.009) + "refugees" (0.009) + "<u>arrive</u>" (0.006) + "spain" (0.006) + "<u>picture</u>" (0.006) + "people" (0.005) + "spanish" (0.005) + "<u>journalists</u>" (0.004) + "sánchez" (0.004) + "spaniards" (0.004) |
| National politics | "immigrants" (0.010) + "<u>spain</u>" (0.009) + "government" (0.008) + "people" (0.007) + "sanchezcastejon" (0.006) + "persons" (0.006) + "politics" (0.005) + "propaganda" (0.004) + "valencia" (0.004) + "years" (0.004) |
| Demanding Government to accept migrants | "psoe" (0.010) + "sanchezcastejon" (0.008) + "immigrants" (0.007) + "spain" (0.007) + "<u>host</u>" (0.006) + "refugees" (0.005) + "circus" (0.005) + "people" (0.004) + "<u>god</u>" (0.004) + "<u>leave</u>" (0.004) |

Source: The authors. For the original Spanish terms, see Table A2 in Appendix A.

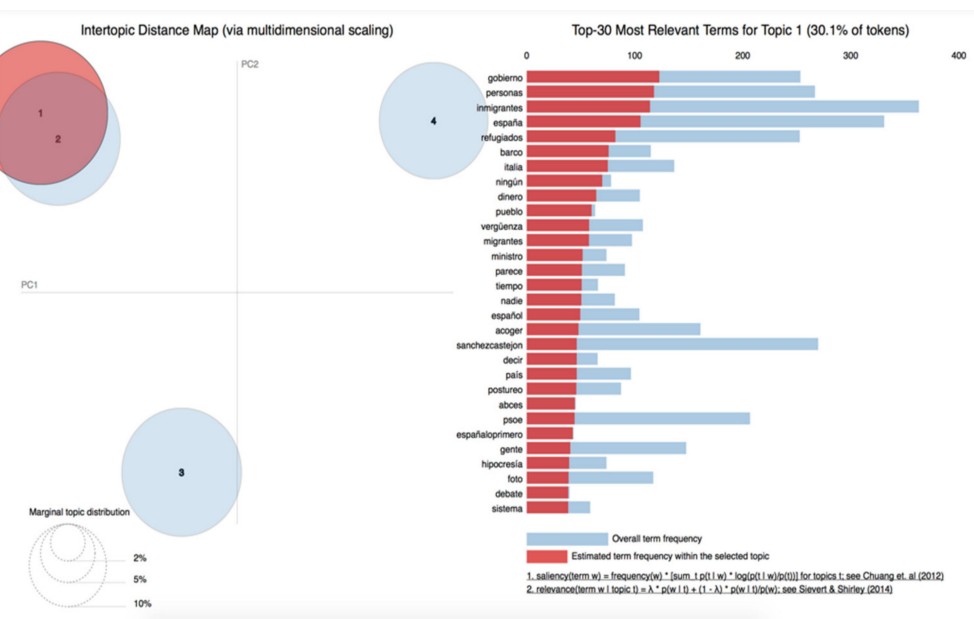

**Figure 3.** Topics within the hateful messages against politicians and list of words of the first topic.

These topics show that, despite the European or, at least, Mediterranean dimension of the event, there is a national perspective in the hateful messages against politicians, and the figures and political parties that are mentioned in these messages seem to be only national. The figures which are targeted in those messages are from different political ideologies and it is not clear whether the attacks come due to having accepted the arrival of the boat or due to not having accepted it earlier. We can see how politicians are rejected from both sides, which makes it impossible to find a clear narrative against politicians, something that was easier to see within the hateful contents against immigrants.

RQ2 and RQ3 allow us to affirm that there is a stronger presence of hate speech contents aimed at politicians than at refugees. However, the values of the sentiment analysis seem to be close in both groups, so it is harder to get a clear image. By comparing the sentiment and the relevance of positive and negative words of those messages with

hateful contents against politicians and against refugees, we can offer more illustrative information. The presence of positive words and the sentiment showed no significant differences, but the average presence of negative words was observed to be higher among those hateful messages aimed at politicians (M = −2.12; SD = 1.341) than at refugees and migrants (M = −2.04; SD = 1.312), [t (5091.164) = −2.437; $p < 0.05$; $d = 0.06$]. The effect is, however, small, so further research would be needed to produce conclusive results in this sense.

Finally, in order to test H1, we analyzed if the hateful contents towards the two studied target groups changed after the announcement of Spanish Prime Minister Pedro Sánchez to receive the boat Aquarius in Spain. This announcement clearly changed the discourse, but not always in the hypothesized direction. Hate speech against refugees significantly increased from M = 0.08 (n = 76; SD = 0.267) before the announcement to M = 0.10 (n = 2320; SD = 0.300) afterwards, [t (1090.294) = −2.576; $p < 0.05$; $d = 0.07$]. The same trend was observed with hate speech against politicians, increasing from M = 0.125 (n = 119; SD = 0.331) to M = 0.173 (n = 4023; SD = 0.378), [t (1094.440) = −4.419; $p < 0.001$; $d = 0.14$], although the effect was slightly bigger in this second case. The differences, that confirm the first part of H1, can be seen more clearly in Figure 4. Nonetheless, the other part of the hypothesis was not confirmed, given that the sentiment of the messages went from a slightly negative one before the announcement (M = −0.162; SD = 1.620) to a slightly positive afterwards (M = 0.026; SD = 1.644), [t (24252) = −3.499; $p < 0.001$; $d = 0.12$]. In this sense, as it can be seen in Figure 5, the presence of positive words significantly increased from M = 1.89 (SD = 1.091) to M = 1.96 (SD = 1.141), [t(1075.918) = −1.996; $p < 0.05$; $d = 0.06$], while the importance of the negative terms significantly decreased from M = −2.05 (SD = 1.334) to M = −1.93 (SD = 1.278), [t(1060.698) = −2.678; $p < 0.01$; $d = 0.09$]. This means that, even though the amount of hate speech against refugees and politicians increased, the language used to discuss the topic became in general less negative after the announcement. Table 4 summarizes the previous information.

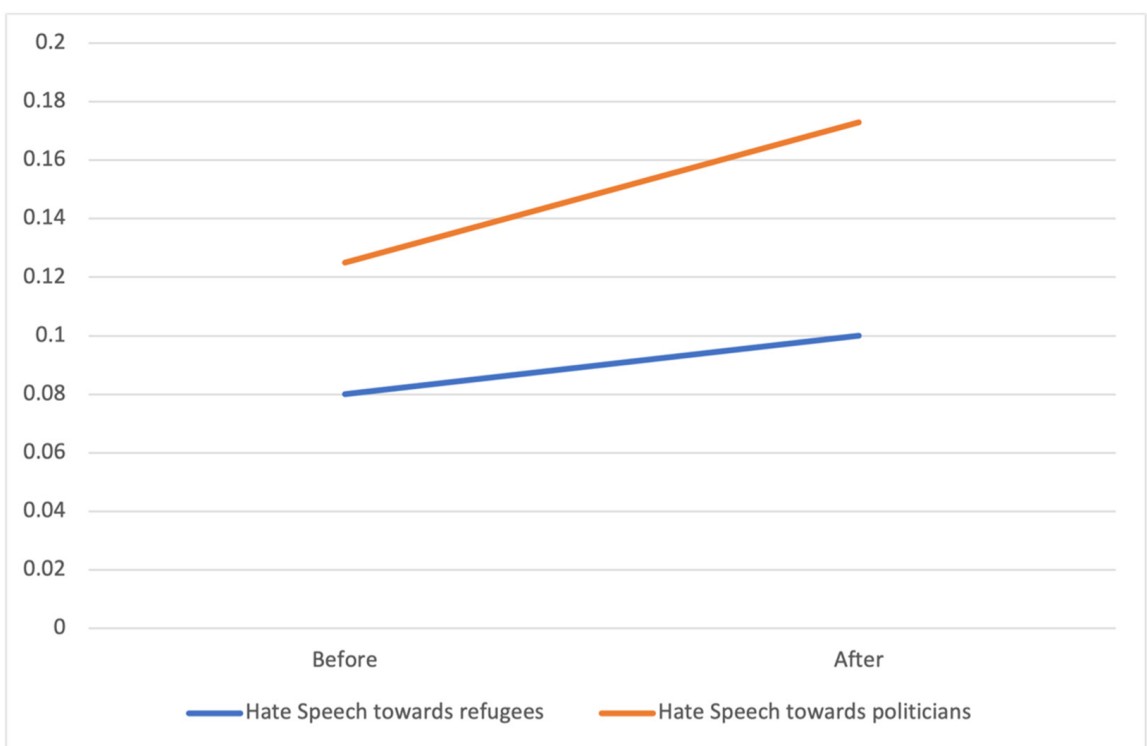

**Figure 4.** Mean values of hate speech against refugees (blue) and politicians (orange) before and after the announcement.

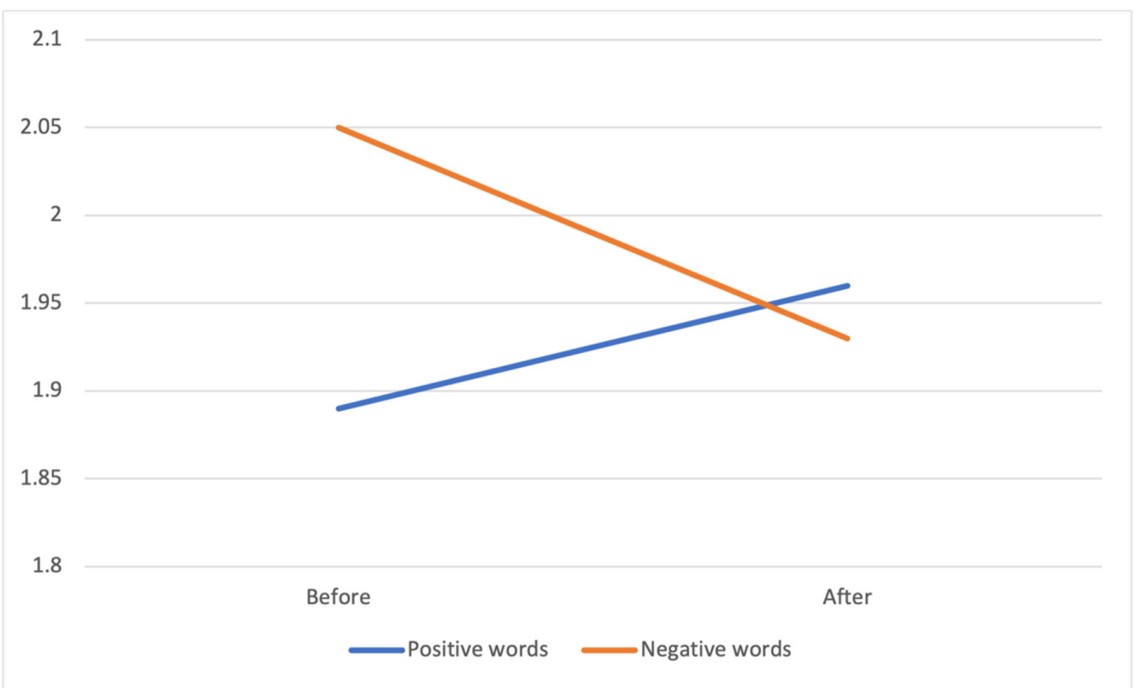

**Figure 5.** Mean values of the presence of positive words (blue) and negative words (orange) before and after the announcement.

**Table 4.** Values before and after the announcement.

|  | Before | | | After | | |
|---|---|---|---|---|---|---|
|  | **n** | **M** | **SD** | **n** | **M** | **SD** |
| Hate Speech towards refugees * | 76 | 0.08 | 0.267 | 2320 | 0.10 | 0.300 |
| Hate Speech towards politicians *** | 119 | 0.125 | 0.331 | 4023 | 0.173 | 0.378 |
| Sentiment *** | - | −0.162 | 1.620 | - | 0.026 | 1.644 |
| Positive words * | - | 1.89 | 1.091 | - | 1.96 | 1.141 |
| Negative words ** | - | −2.05 | 1.334 | - | −1.93 | 1.278 |

Source: The authors. * = $p < 0.05$; ** = $p < 0.01$; *** = $p < 0.001$.

## 4. Discussion and Conclusions

The study has shown that a relevant part of the conversation around a high-profile event such as the arrival of the Aquarius boat to Spain included hateful or rejection expressions against refugees and migrants and, to a greater extent, against politicians. The sentiment of the language used in this conversation was, in general, positive, although the presence of hate speech and the sentiment showed a correlation, making the sentiment more negative in those messages expressing hate or rejection. For this reason, the tweets with hate speech, both against refugees and against politicians were more negative, with a stronger relevance of negative words, while the positive ones remained rather similar in all kinds of messages. In addition, although there was significantly more hate against politicians than against refugees and migrants, the sentiment of the language in both cases remained quite similar, and only the relevance of negative words was slightly bigger among the tweets expressing hate against politicians. These initial findings support the natural perception of conceiving hate speech as a product of negative feelings expressed in language.

With all these observations, we can conclude that the conversation around the boat Aquarius in Twitter in Spain was not predominantly hateful or negative –something that might be related with the more empowering discourse used in television coverage [56], and that politicians were most commonly criticized or attacked than migrants. The differences, however, were not big in any case. Future studies analyzing similar events—in summer 2019 a similar situation was seen in the Mediterranean with a boat of the Open Arms NGO—

will be able to test if the conversation in Twitter has changed since this first "influential" rescue boat.

The unsupervised analysis of the topics discovered that hate speech against refugees usually revolves around the potential pull effect that hosting the boat would have, as well as the threats that immigrants could pose—terrorism or economic burdens for the State. This shows the relevant role played by stereotypes and prejudices, if not complete lies or fake news, such as the connection of immigration with terrorism. This is coherent with the observations made in previous studies [15,17,20], and highlights the need for initiatives and studies that approach the interaction between hate speech and disinformation [31,44]. In general, the topics associated to hate speech around the studied event are strongly connected with the frames observed by Lawlor and Tolley [57] in the media representation of immigration.

These frames are also perceived in the case of the hateful messages aimed at politicians, in which topics focused more specifically on the shameful behavior of these officials, both for hosting and for not hosting the asylum seekers, usually with a stronger presence of national politics rather than an international one. This element of the study, although preliminary, can show how hateful messages aimed at politicians have a clear ideological background, which constitutes one of the types of hate typified by the Statistic System of Criminality (SEC) of the Ministry of the Interior of Spain, and which is strongly connected with affective polarization [58].

Finally, we could observe that the presence of hate speech against both target groups significantly increased after the announcement of the Spanish Prime Minister of welcoming the boat in Valencia's harbor, which confirmed our main hypothesis. On the other hand, the sentiment of the language used in the discussion became more positive, with a stronger presence of positive words and a weaker presence of negative ones, something that theoretically clashes with the increase in the presence of hateful expressions. This shows how, despite the correlation between language sentiment and presence of hate, it is not determinant and further analyses are needed in order to really establish the connection between these two elements.

This paper acknowledges some limitations, including the fact that the results only covered 9 days of conversation. Even though they were the most active days, covering the decision of hosting the boat and its arrival to Spain, a broader analysis in terms of time and of sample size could have strengthened the conclusions. The size of the sample also made it impossible to conduct comparative topic modelling for each group before and after the reception, something that could be done with the sentiment analysis, and that could have been interesting in order to observe potential changes in the mains topics before and after the reception of the boat. Secondly, the automatic sentiment analysis is based on an *a priori lexicon* that, although validated, was generated in a different context. Additionally, recent research shows that for short texts (such as tweets), a supervised sentiment analysis may produce better results than an automatic sentiment analysis based on dictionaries [59], which implies that this approach should be taken into account in future studies. In the same line, topic modelling tends to be used in larger texts, such as articles from newspapers [60], but there are some arguments that pushed us to employ it in tweets: The longer extension of tweets since 2017—280 characters rather then 140, the relevance of Twitter in the construction of public discourse in the present, and the interest to test this technique in this medium in Spain, discovering whether it can be applied in larger studies. This helps increase the corpus of works employing topic modelling as a text mining research technique on Twitter, in the line with the observations of Karami et al. [17], and it serves as a basis for future studies in the same line in the Spanish setting.

This way, the study offers preliminary observations that need further analysis and confirmation from future studies, as well as an update of the predominant sentiments and topics, given that, despite the milestone condition of the arrival of the Aquarius boat, the conversation around the topic of asylum and migration might have changed significantly, especially after the appearance of a right-wing anti-immigration party such as Vox [15].

However, the study offers promising preliminary conclusions on the relationship between hate speech and sentiments, using novel techniques that have rarely been exploited in migration research, and especially gives empirical evidence of how milestones during high-profile events can change the way people discuss public opinion topics in social media.

**Author Contributions:** C.A.-C. designed the study and collected the data; D.B.-H. and M.F.-V. conducted data analysis and wrote the report; F.S. contributed to the rationale and discussion of results. All authors have read and agreed to the published version of the manuscript.

**Funding:** This research was funded by the European Union's Right Equality and Citizenship Programme (2014–2020). REC-RRAC-RACI-AG-2019 grant agreement 875217.

**Institutional Review Board Statement:** Not applicable.

**Informed Consent Statement:** Not applicable.

**Data Availability Statement:** The data presented in this study are available on request from the corresponding author. The data are not publicly available due to their restricted access only using Twitter's API.

**Conflicts of Interest:** The authors declare no conflict of interest.

## Appendix A

**Table A1.** Topics of the messages with hate speech against refugees in Spanish.

| Topic | Most Common Words and Their Frequency within the Topic (the Underlined Words are the Most Determinant for the Labelling of the Topic) |
|---|---|
| Pull effect and consequences | "inmigrantes" (0.008) + "españa" (0.007) + "barco" (0.005) + "efecto" (0.005) + "llamada" (0.004) + "va" (0.004) + "valencia" (0.004) + "consecuencias" (0.004) + "europa" (0.004) + "concentración" (0.004) |
| Pull effect and not welcoming "illegals" | "inmigrantes" (0.013) + "personas" (0.008) + "españa" (0.008) + "llamada" (0.007) + "efecto" (0.007) + "ilegales" (0.007) + "defiendetusfronteras" (0.007) + "españoles" (0.006) + "puerto" (0.006) + "aquariusnotwelcome" (0.006) |
| Not welcoming and terrorism | "van" (0.008) + "españa" (0.008) + "personas" (0.006) + "aquariusnotwelcome" (0.005) + "país" (0.005) + "boko" (0.005) + "haram" (0.005) + "inmigrantes" (0.005) + "barco" (0.004) + "tener" (0.004) |
| Smugglers and NGOs | "inmigrantes" (0.023) + "españa" (0.010) + "españoles" + "vienen" (0.008) + "mafias" (0.008) + "personas" (0.007) + "van" (0.006) + "ilegales" (0.006) + "ongs" (0.006) + "barco" (0.006) |
| Money and jobs | "refugiados" (0.011) + "inmigrantes" (0.009) + "pagar" (0.007) + "españa" (0.007) + "países" (0.005) + "barco" (0.005) + "españoles" (0.005) + "personas" (0.005) + "solución" (0.005) + "trabajo" (0.005) |
| Entrance to Europe | "españa" (0.018) + "inmigrantes" (0.009) + "europa" (0.006) + "barco" (0.006) + "valencia" (0.005) + "españoles" (0.005) + "gobierno" (0.005) + "inmigración" (0.005) + "norte" (0.004) + "millones" (0.005) |

Source: The authors.

**Table A2.** Topics of the messages with hate speech against politicians in Spanish.

| Topic | Most Common Words and Their Frequency within the Topic (the Underlined Words are the Most Determinant for the Labelling of the Topic) |
|---|---|
| Shame of politicians | "inmigrantes" (0.006) + "urdangarín" (0.005) + "personas" (0.005) + "europa" (0.005) + "gobierno" (0.004) + "pp" (0.004) + "italia" (0.004) + "vergüenza" (0.004) + "hipocresía" (0.004) + "abierto" (0.003) |
| Media coverage | "valencia" (0.009) + "refugiados" (0.009) + "llegan" (0.006) + "españa" (0.006) + "foto" (0.006) + "personas" (0.005) + "español" (0.005) + "periodistas" (0.004) + "sánchez" (0.004) + "españoles" (0.004) |
| National politics | "inmigrantes" (0.010) + "españa" (0.009) + "gobierno" (0.008) + "gente" (0.007) + "sanchezcastejon" (0.006) + "personas" (0.006) + "política" (0.005) + "propaganda" (0.004) + "valencia" (0.004) + "años" (0.004) |
| Demanding Government to accept migrants | "psoe" (0.010) + "sanchezcastejon" (0.008) + "inmigrantes" (0.007) + "españa" (0.007) + "acoger" (0.006) + "refugiados" (0.005) + "circo" (0.005) + "personas" (0.004) + "dios" (0.004) + "dejo" (0.004) |

Source: The authors.

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
