# Peer review of "Refugees Welcome? Online Hate Speech and Sentiments in Twitter in Spain during the Reception of the Boat Aquarius"

_sustainability, doi:10.3390/su13052728_

Round 1

Reviewer 1 Report

Authors aim to provide an analysis of Twitter comments regarding politicians and refugees during the reception of the refugee boat. The main flaw of the paper is that it presents mostly the tweet analysis in bulk, for the whole period together. However, it would be useful if the topic and phrases extraction would change before and after the reception of the refugee boat. Besides, the paper does not reflect that authors have in-depth knowledge about the text mining research on Twitter papers.

  1. Overall - the paper does not reflect that authors have in-depth knowledge about the text mining research on Twitter papers. Authors should find relevant papers on Twitter and text mining, and text mining and social media in general and read them. Based on that, they should correct introduction, literature review and conclusions.
  2. Authors do not present an elaboration about the theoretical and practical contributions in the introduction and conclusion, concerning previous research.
  3. Authors should reconsider to explain the section about the scientific contribution in the introduction, as well as in the conclusion part of the paper, with the structured comparison of the current research with previous research. The text can be one paragraph long, but it should contain the most important studies.
  4. Authors mention that they provided a manual analysis of three measures in 24,254 messages: data and time, presence of hateful expressions, and the target of hateful expression. It seems to me that this amount of tweets would require a huge amount of time for this analysis. In addition, the validity of the coding is not elaborated, in terms of a number of coders.
  5. Topic extraction is not clearly elaborated regarding the internal coherence valued. It should be better elaborated about how topics were selected, both in terms of terms selection and in terms of a number of topics.
  6. Chapter 2.3 Analysis should be renamed into Software used. In this chapter, possible problems with software and data preparation should be elaborated.
  7. Figure 1 is unclear! This is likely some mistake. Graphs are missing x and y-axis. Numbers are too small.
  8. Title of Table 2 is likely a mistake. Text in Table 2 should be translated into English. Possibly, tables in Spanish could be removed to the Appendix, if authors’ opinion is that it would be useful. The same refers to Table 3.
  9. Figure 2 is too small. It should be divided into two figures, and each should be presented with much better quality. The same refers to Figure 3, 4 and 5.
  10. Authors mention H1 on pages 10 and 11. However, hypothesis is not mentioned in previous parts of the paper.
  11. Conduct the analysis in Table 2 and 3 separately before and after the reception of the refugee boat.
  12. Future research directions are trivial.

Literature suggestions:

Pejić Bach, M., Krstić, Ž., Seljan, S., & Turulja, L. (2019). Text mining for big data analysis in financial sector: A literature review. Sustainability11(5), 1277.

Pejić Bach, M., Pulido, C. M., Suša Vugec, D., Ionescu, V., Redondo-Sama, G., & Ruiz-Eugenio, L. (2020). Fostering Social Project Impact with Twitter: Current Usage and Perspectives. Sustainability12(15), 6290.

Author Response

Authors aim to provide an analysis of Twitter comments regarding politicians and refugees during the reception of the refugee boat. The main flaw of the paper is that it presents mostly the tweet analysis in bulk, for the whole period together. However, it would be useful if the topic and phrases extraction would change before and after the reception of the refugee boat. Besides, the paper does not reflect that authors have in-depth knowledge about the text mining research on Twitter papers.

  1. Overall - the paper does not reflect that authors have in-depth knowledge about the text mining research on Twitter papers. Authors should find relevant papers on Twitter and text mining, and text mining and social media in general and read them. Based on that, they should correct introduction, literature review and conclusions.

A paragraph about text mining has been added after having reviewed more literature about the topic, and a total of 7 new references have been added. All this can be found in lines 67-77 in the introduction and in lines 521-523 in the conclusions.

  1. Authors do not present an elaboration about the theoretical and practical contributions in the introduction and conclusion, concerning previous research.

This has been included in lines 136-151, where we explain the novelty of the study and its theoretical contribution; we have also included three new references on the topic. The discussion with previous studies is also conducted in lines 481-486 of section 4, and more practical implication of the study are suggested in lines 491-494 and in lines 521-523.

  1. Authors should reconsider to explain the section about the scientific contribution in the introduction, as well as in the conclusion part of the paper, with the structured comparison of the current research with previous research. The text can be one paragraph long, but it should contain the most important studies.

Previous research has been added to the discussion, and it has been specified what is the knowledge gap that it fills in. This can be found, together with the contents of the previous suggestion, in lines 136-151of the introduction and in lines 481-486 of the conclusion.

  1. Authors mention that they provided a manual analysis of three measures in 24,254 messages: data and time, presence of hateful expressions, and the target of hateful expression. It seems to me that this amount of tweets would require a huge amount of time for this analysis. In addition, the validity of the coding is not elaborated, in terms of a number of coders.

We have added the information about the duration of the manual analysis and we have clarified the validity of the coding, highlighting the number of coders and the inter-coders measures; this can be found in lines 269-276. We would like to add that data and time did not require coding, given that they are provided by the downloaded contents in JSON format.

  1. Topic extraction is not clearly elaborated regarding the internal coherence valued. It should be better elaborated about how topics were selected, both in terms of terms selection and in terms of a number of topics.

The process of topic extraction has been more clearly explained, with more detail about the internal coherence values that made us select those numbers of topics, about the reason for selecting that number of terms in each topic, and about the reason for choosing this model of selecting terms rather than others. This information can be found in lines 297-305, as well as in footnotes 3 and 4 and in Table 2.

  1. Chapter 2.3 Analysis should be renamed into Software used. In this chapter, possible problems with software and data preparation should be elaborated.

We have renamed this section and we have further explained the process and the software used during the analysis. This can be found in lines 323-335.

  1. Figure 1 is unclear! This is likely some mistake. Graphs are missing x and y-axis. Numbers are too small.

A new figure, with higher resolution has been added. For that, a new figure had to be created, given that the model is unsupervised, and every time the data are newly run randomly change, although maintaining the trend. A clearer interpretation of what information appear in each axis can be found in lines 299-301. Addittionally, the number of topics and internal coherence values have been further explained, in lines 301-305.

  1. Title of Table 2 is likely a mistake. Text in Table 2 should be translated into English. Possibly, tables in Spanish could be removed to the Appendix, if authors’ opinion is that it would be useful. The same refers to Table 3.

The title has been corrected; it was indeed, a mistake. Following the suggestion of the reviewer, we have translated the tables and put the Spanish versions in Appendix 1.

  1. Figure 2 is too small. It should be divided into two figures, and each should be presented with much better quality. The same refers to Figure 3, 4 and 5.

All figures have been made bigger for clarity. They have not been divided into two because content from the map and the list of words overlaps and it should be interpreted together.

  1. Authors mention H1 on pages 10 and 11. However, hypothesis is not mentioned in previous parts of the paper.

Epigraph 1.1.2. justifies our H1, which is answered with the results from line 424 until the end of the results section, and it is also referred again in the discussion section in lines 495-503. In line 94, at the end of the introduction, it is mentioned that we will justify our RQs and our hypothesis in the following sections; in order to clarify this aspect, the section in which is aspect is treated has been added.

  1. Conduct the analysis in Table 2 and 3 separately before and after the reception of the refugee boat.

The analysis cannot be conducted, given that the number of hateful tweets collected before the reception of the boat –76 against refugees and 119 against politicians– were not enough for this kind of analysis. The number of hateful tweets before and after the reception have been added to Table 4, so this information can be included, and the absence of this analysis has been mentioned as a limitation in lines 507-511.

  1. Future research directions are trivial.

More specific suggestions for future research have been provided in lines 521-523 and 525-528.

Literature suggestions:

Pejić Bach, M., Krstić, Ž., Seljan, S., & Turulja, L. (2019). Text mining for big data analysis in financial sector: A literature review. Sustainability11(5), 1277.

Pejić Bach, M., Pulido, C. M., Suša Vugec, D., Ionescu, V., Redondo-Sama, G., & Ruiz-Eugenio, L. (2020). Fostering Social Project Impact with Twitter: Current Usage and Perspectives. Sustainability12(15), 6290.

We appreciate the suggestions; both of them have been read, together with other works, and included in the paragraph about text mining in line 72.

Reviewer 2 Report

Overall, his is a powerful and well-grounded piece. It is very relevan given he current situation around public debates and discussions on refugees in Spain and the effects that the economic downturn (because of COVID) will have on public views on immigration in more general terms. The methodology is sound and the argument well-structured. It need a revision of spelling. 

Author Response

Overall, his is a powerful and well-grounded piece. It is very relevan given he current situation around public debates and discussions on refugees in Spain and the effects that the economic downturn (because of COVID) will have on public views on immigration in more general terms. The methodology is sound and the argument well-structured. It need a revision of spelling. 

We thank the reviewer for his/her comments. A spelling revision has been conducted along the whole text.

Reviewer 3 Report

The study offers promising preliminary conclusions about relationship between hate speech and sentiments, using novel techniques that have rarely been exploited in migration research, and gives empirical evidence of how mile-stones during high-profile events can change the way people discuss public opinion topics in social media.

Author Response

The study offers promising preliminary conclusions about relationship between hate speech and sentiments, using novel techniques that have rarely been exploited in migration research, and gives empirical evidence of how mile-stones during high-profile events can change the way people discuss public opinion topics in social media.

We thank the reviewer for his/her comments

Round 2

Reviewer 1 Report

Dear authors, thank you for your hard work. To my opinion, the paper has been substantially improved and is now ready for publication.